# Influence of Body Mass Index on Gestation and Delivery in Nulliparous Women: A Cohort Study

**DOI:** 10.3390/ijerph16112015

**Published:** 2019-06-06

**Authors:** Noemí Rodríguez-Mesa, Paula Robles-Benayas, Yolanda Rodríguez-López, Eva María Pérez-Fernández, Ana Isabel Cobo-Cuenca

**Affiliations:** 1Hospital Virgen de la Salud de Toledo, Servicio de salud de Castilla la Mancha (SESCAM), 45005 Toledo, Spain; noemirodriguezmesa@gmail.com (N.R.-M.); benayitas449@gmail.com (P.R.-B.); yolandarl_87@yahoo.es (Y.R.-L.); evamap01@ucm.es (E.M.P.-F.); 2Department of Nursing and Physiotherapy, Grupo Investigación multidisciplinar en Cuidados (IMCU), Universidad de Castilla la Mancha, 45004 Toledo, Spain

**Keywords:** obesity, maternal body mass index, labor, nulliparous women

## Abstract

*Aims*: To assess the influence of obesity on pregnancy and delivery in pregnant nulliparous women. Methods: A cohort, longitudinal, retrospective study was conducted in Spain with 710 women, of which 109 were obese (BMI > 30) and 601 were normoweight (BMI < 25). Consecutive nonrandom sampling. *Variables*: maternal age, BMI, gestational age, fetal position, start of labor, dilation and expulsion times, type of delivery and newborn weight and height. Results: The dilation time in obese women (309.81 ± 150.42 min) was longer than that in normoweight women (281.18 ± 136.90 min) (*p* = 0.05, Student’s *t*-test). A higher fetal weight was more likely to lead to longer dilation time (OR = 0.43, 95% CI 0.010–0.075, *p* < 0.001) and expulsion time (OR = 0.027, 95% CI 0.015–0.039, *p* < 0.001). A higher maternal age was more likely to lead to a longer expulsion time (OR = 2.054, 95% CI 1.17–2.99, *p* < 0.001). Obese women were more likely to have gestational diabetes [relative risk (RR) = 3.612, 95% CI 2.102–6.207, *p* < 0.001], preeclampsia (RR = 5.514, 95% CI 1.128–26.96, *p* = 0.05), induced birth (RR = 1.26, 95% CI 1.06–1.50, *p* = 0.017) and cesarean section (RR = 2.16, 95% CI 1.11–4.20, *p* = 0.022) than normoweight women. *Conclusion*: Obesity is associated with increased complications during pregnancy, an increased incidence of a cesarean section and induced birth but it has no significant effect on the delivery time.

## 1. Introduction

The prevalence of obesity has increased in recent years worldwide. It is estimated that by 2025, the global prevalence of obesity in women will be greater than 21% [1]. Obesity affects 17% of the adult population in Spain. In Castilla-La Mancha, more than 18% of the female population are obese [2].

Obesity is considered a risk factor for several diseases and is related to many pathologies, including cardiovascular diseases, type 2 diabetes, hypertension, kidney problems and cancer, such as breast, colon and endometrial cancer [3,4].

Obesity has increased among pregnant women in recent decades, resulting in maternal, perinatal and neonatal complications. A higher body mass index (BMI) results in a greater risk of gestational and fetal pathology [5,6,7,8]. Obese pregnant women have higher risks of gestational diabetes, pregnancy hypertension, preeclampsia and HELLP syndrome [9,10,11]. The fetal risks associated with maternal obesity are preterm birth, fetal macrosomia, perinatal death and congenital anomalies [5].

Obesity results in an increased duration of delivery, increased rates of instrumental deliveries and cesarean sections, as wells as in risk of a postpartum hemorrhage, perianal tears, complications from anesthesia and surgical procedures, surgical wound infections and urinary tract infections [11,12,13]. The associated pathologies and fetal risks favor the induction of birth [12,13,14].

Morbidity and mortality associated with excess weight in pregnant women have been studied extensively in recent years; however, the effects of obesity during childbirth have not been examined. Currently, delivery rooms in Spain attend to a large number of obese women. These pregnant women exhibit different characteristics during delivery, which makes it necessary to investigate the time required for different stages of birth. To our knowledge, in Spain there have been no studies of nulliparous women with obesity and the relationships between obesity and dilation and expulsion times. 

The aims of this study were: (1) to assess the influence of obesity on pregnancy and childbirth; (2) to determine the relationship between BMI and dilation and expulsion times in nulliparous women; and (3) to identify other variables that may influence the duration of childbirth.

## 2. Materials and Methods

### 2.1. Study Design and Participants

This was a quantitative, retrospective, longitudinal cohort study conducted in the delivery unit of a tertiary hospital of reference in Spain.

Through consecutive nonprobabilistic sampling, 710 nulliparous women who went into labor during 2014–2015 and met the inclusion and exclusion criteria were selected. Two cohorts were created within this group:Group 1 (obese women) [15]: cohort of 601 pregnant women with a BMI > 30 kg/m^2^;Group 2 (normoweight women) [15]: cohort of 109 pregnant women with a BMI between 18.5 and 24.9 kg/m^2^.

The inclusion criteria were as follows: nulliparous pregnant women with a single term fetus (37–42 weeks), fetal cephalic presentation and epidural analgesia.

The exclusion criteria were as follows: scheduled cesarean delivery, overweight women, multiparous women, placenta previa and/or accreta, multiple gestation, noncephalic fetal presentation, intrauterine fetal death and stained amniotic fluid. 

The researchers collected data using the childbirth records belonging to the hospital. This register contained all information necessary for this study. Maternal weight and height of the women were recorded at the first prenatal consultation with a midwife. In Spain, prenatal control is free for all women and the first visit to a midwife is at 8 weeks of gestation.

### 2.2. Sources and Information

The study variables included: (1) maternal variables: age, weight, height, BMI, gestational diabetes and preeclampsia; (2) obstetric variables: type of labor (spontaneous or induced), duration of dilation and expulsion in minutes, type of delivery (vaginal delivery, instrumental (forceps/ventouse) delivery and cesarean delivery), amniorrhexis (spontaneous or artificial) and the use of oxytocin; and (3) fetal variables: gestational age, fetal cephalic presentation and newborn weight and height.

The sample size was calculated using the Granmo program (version 7.12) (Barcelona, Spain, 2012), based on an α risk of 0.05, a relative risk (RR) of 1.7, a β risk of 0.2, an obesity percentage of 18%, a cesarean rate in the normoweight group of 0.2 and a replacement rate of 10%. These factors indicated that a population of 610 subjects in the normoweight group and 110 subjects in the obesity group would be necessary for the study [16].

After approval by the ethics and clinical research committee (Comité de Ética del Complejo hospitalario de Toledo, dossier number CEIC TO-114/2014) of the institution, data were collected retrospectively, starting with the most recent cases until the sample needed to conduct the study was complete.

Statistical analysis was performed with the statistical program SPSS v.19 (IBM, Armonk, NY, USA). Percentages, means and dispersion measures were obtained for the descriptive analysis. Chi-squared (χ^2^; qualitative) inferential analysis, Student’s *t*-test, Fisher’s test (quantitative) and Spearman’s correlation coefficient analysis (quantitative) were performed. Multivariate linear regression model was used with quantitative variables. The effect of the BMI category (normoweight or obese) on each clinical outcome was assessed using a binomial regression model. The results are presented as RR with a 95% confidence interval (CI), assuming bilateral significance indicated by *p* < 0.05.

## 3. Results

The sample consisted of 720 women, of which ten were eliminated for an extreme age (three < 18 years old and seven > 40 years). The definitive sample consisted of 710 women, aged between 18 and 40 years, with an average age of 30.06 ± 5.09 years. Overall, 6.7% (48) of the participants had gestational diabetes and 0.8% (6) had preeclampsia.

Among the participants, 601 (84.6%) were normoweight (BMI < 25) and were included in group 2. The remaining 109 participants (15.4%) were obese (BMI > 30) and were included in group 1. Group 1 (obese) included significantly more participants with gestational diabetes (*p* < 0.001, χ^2^) and preeclampsia (*p* = 0.05, Fisher’s test) than did group 2 (normoweight) (Table 1).

The obese women were 3.61 times more likely to have gestational diabetes (RR 3.61, 95% CI 2.1–6.2, *p* < 0.001) and 5.5 times more likely to have preeclampsia (RR 5.51, 95% CI 1.13–26.96, *p* = 0.05) than were the normoweight women (Table 2).

Group 1 (obese) included more participants who underwent induction (*p* = 0.017, χ^2^) and a cesarean section (*p* = 0.018, χ^2^) than did group 2 (normoweight) (Table 3).

The obese women were 1.26 times more likely to have induced labor (RR 1.26, 95% CI 1.06–1.50, *p* = 0.017) and 2.1 times more likely to have a cesarean section (RR 2.16, 95% CI 1.11–4.22, *p* = 0.022) than were normoweight women (Table 2).

### 3.1. First Stage of Labor

The dilation time (first stage of labor) was longer (*p* = 0.05, Student’s *t*-test) in group 1 (mean ± standard deviation: 309.81 ± 150.42 min) than in group 2 (281.18 ± 136.90 min) (Table 3).

The dilation time (first stage of labor) was significantly related to the birth weight (Spearman’s R = 0.169, *p* < 0.001) and height (Spearman’s R = 0.160; *p* < 0.001) of the child. The dilation time (first stage of labor) was significantly related to the birth weight (*p* < 0.001, Student’s-*t*-test) and fetal weight ≥4000 g (375.5 ± 152.99 g) vs. fetal weight <4000 g (281.53 ± 137.45 g). The time of the first stage of labor with the administration of oxytocin (304.70 ± 142.28 min) was significantly longer (*p* < 0.001, Student’s *t*-test) than that without oxytocin (211.92 ± 94.00 min). No significant relationships were found among the dilation time, labor onset type and position of the head of the fetus.

### 3.2. Second Stage of Labor

The expulsion time in older women (30–40 years old; 112.57 ± 64.49 min) was significantly longer (*p* < 0.001, Student’s *t*-test) than that in younger women (18–29 years old; 96.46 ± 65.65 min).

The expulsion time was significantly longer (*p* = 0.011, Student’s *t*-test) when the fetal head was in the occiput posterior position (129.39 ± 64.77 min) than when it was in the occiput anterior position (105.83 ± 64.60 min).

No differences were found in the expulsion time depending on the administration of oxytocin, BMI, rupture of membranes and onset of labor.

The weight (Spearman’s R = 0.188, *p* < 0.001) and height (Spearman’s R = 0.106, *p* = 0.005) of the child and the age of the mother (Spearman’s R = 0.159, *p* < 0.001) exhibited significant weak positive relationships with the expulsion time.

Table 4 shows factors associated with the dilation and expulsion times. 

## 4. Discussion

The results of our study suggest that obesity in primiparous pregnant women is associated with the occurrence of complications during pregnancy (diabetes and preeclampsia) and a greater number of complications during delivery (greater numbers of inductions, cesarean sections or instrument-assisted births and longer duration of dilation and delivery).

Consistent with the findings of other studies, the obese group had a significantly higher incidence of gestational diabetes [13,17] and preeclampsia than did the normoweight group [18].

In addition, the incidences of instrumental births and cesarean sections were higher in the obese group [11,12,18,19,20,21,22]. Obese women were more likely to have induced births [12,13,20,21] and a longer duration of the first stage of labor than were normoweight women [10,12,21,22]. This increase in the time of dilation in obese women may be due to uterine dystocia [23,24], such as cephalopelvic disproportion [25].

A higher fetal weight is associated with a longer duration of the first stage of labor [25]. Among the factors that are associated with a greater weight and height of the newborn, maternal obesity [11,17], weight gain and diabetes are noteworthy [17,26,27]. In our study, obese women were 2.7 times more likely to have children weighing more than 4000 g. Although the obese group included a higher proportion of diabetic women [11,17], we did not find a relationship between gestational diabetes and the fetal weight greater than 4000 g. This finding may have been due to the fact that Spain has a public health system and when gestational diabetes is detected, women undergo health consultations. The weight and height of the child are also related to the expulsion time.

Controversy exists regarding the relationship between obesity and the expulsion time. Carhall et al. [11] found a shorter expulsion time in obese women. Other studies found a direct relationship between the dilation time and expulsion time, in the absence of other factors [25]. In this study, we did not find differences in the expulsion time, possibly because at our center, the expulsion time is not allowed to be longer than 3 h, as established by the hospital protocol. 

A longer expulsion time is related to the occiput posterior position [28] and an older maternal age [29].

In this study, the dilation period was longer in women who received oxytocin. This finding contradicts to the data of other studies [30,31], possibly because the administration of oxytocin is conducted in a generalized manner. Thus, deliveries with an excessively short or considerably shorter dilation phase than the average phase in nulliparous women or than that expected by professionals under different circumstances did not include this intervention.

There are limitations in this study. Because a cohort study design was used, it was not possible to establish causal relations. Furthermore, the R values were very weak and thus, it could be interesting to repeat this study, with a larger sample, to evaluate the association between the age and the time of delivery. The data collected were obtained from the birth register of the hospital. The other limitation was that we did not collect data about the weight gain during pregnancy.

The strength of this study is that a power calculation was performed, and an appropriate number of cases was included in each group. 

This study did not receive any specific grant from funding agencies in the public, commercial or not-for-profit sectors.

## 5. Conclusions

Obesity in pregnant nulliparous women is associated with numerous risks during pregnancy, such as gestational diabetes and preeclampsia.

In addition, obesity is associated with an increased rate of labor induction, a higher probability of cesarean section, a longer dilation phase and an increased risk of the birth weight >4000 g.

A higher fetal weight is associated with longer dilation and expulsion times.

An increased expulsion time is associated with an older maternal age.

### Implications for Practice

Owing to the influence of excessive weight gain in pregnant women, it is important to focus on weight control, from gestation to the delivery period, to reduce the occurrence of complications and prolonged delivery times and to facilitate recovery.

## Figures and Tables

**Table 1 ijerph-16-02015-t001:** Maternal and fetal variables in the normoweight and obesity groups.

Variables	Group 1 Obesity (BMI > 30 kg/m^2^) (*n* = 109)	Group 2 Normoweight (BMI < 25 kg/m^2^) (*n* = 601)	*p*
Maternal variables			
Age	29.84 ± 5.23	30.1 ± 5.06	ns
Age range			
18–29 years old	52 (18.1%)	235 (81.9%)	ns
30–40 years old	57 (13.5%)	366 (86.5%)	
Weight (kg)	88.36 ± 11.07	58.78 ± 6.25	<0.001
Body mass index (BMI; kg/m^2^)	33.6 ± 3.62	22.0 ± 2.14	<0.001
Diabetes	19 (17.4%)	29 (4.8%)	<0.001
Preeclampsia	3 (2.8%)	3 (0.5%)	0.05 *
Fetal variables			
Fetal age (weeks)	39.53 ± 1.29	39.53 ± 1.95	ns
Fetal weight (g)	3363 ± 523.94	3250 ± 383.9	0.034
Numbers of children with fetal weight ≥4000 g	10 (9.2%)	20 (3.3%)	0.05
Fetal height (cm)	49.96 ± 1.69	49.73 ± 1.87	ns

* Fisher’s test (<5); ns: not significance.

**Table 2 ijerph-16-02015-t002:** Effects of maternal body mass index on pregnancy outcomes.

Outcome	RR * (95% CI ^†^)	*p*-Value
Diabetes		
Obese vs. normoweight [19 (17.4%) vs. 29 (4.8%)]	3.61 (2.1–6.2)	<0.001
Preeclampsia		
Obese vs. normoweight [3 (2.7%) vs. 3 (0.5%)]	5.51 (1.13–26.96)	0.05
Cesarean delivery		
Obese vs. normoweight [11 (10.1%) vs. 28 (4.7%)]	2.16 (1.11–4.20)	0.022
Induction of labor		
Obese vs. normoweight [65 (59.6%) vs. 284 (47.3%)]	1.26(1.06–1.50)	0.017
Weight birth ≥4000 g		
Obese vs. normoweight [10 (9.2%) vs. 20 (3.3%)]	2.75 (1.33–5.73)	0.005

* Relative risk: the treatment effects were presented as relative risks (95% CI) using a log binomial model that included the body mass index. ^†^ CI: confidence interval. Fisher’s test was used for preeclampsia.

**Table 3 ijerph-16-02015-t003:** Obstetric variables in the normoweight and obesity groups.

Obstetric Variables	Group 1 Obesity (*n* = 109) (BMI * > 30 kg/m^2^)	Group 2 Normoweight (*n* = 601) (BMI * < 25 kg/m^2^)	*p*
Mode of delivery			
Spontaneous/vaginal birth	83 (76.1%)	441 (73.4%)	
Forceps/ventouse (instrumental)	15 (13.8%)	132 (22%)	0.018
Cesarean	11 (10.1%)	28 (4.7%)	
Labor			
Spontaneous	44 (40.4%)	17 (52.7%)	
Induced	65 (59.6%)	284 (47.3%)	0.017
Rupture of membranes			
Spontaneous	48 (44%)	265 (44.2%)	ns
Induced	61 (56%)	335 (55.8%)	
Oxytocin			
No	20 (18.3%)	124(20.6%)	
Yes	89(81.7%)	477 (79.4%)	ns
	M ± DT	M ± DT	
Length of first stage of labor	309.81 ± 150.42	281.18 ± 136.9	0.05
Length of second stage of labor	109.9 ± 65.05	105.39 ± 65.48	ns
Length of labor	409.75 ± 165.26	384.89 ± 152.8	ns

* BMI, body mass index. ns: not significance.

**Table 4 ijerph-16-02015-t004:** Factors associated with the dilation and expulsion times.

Dilation Time	OR * (95% CI^†^)	*p*-Value	Expulsion Time	OR * (95% CI^†^)	*p*-Value
Maternal age	―	ns	Maternal age	2.052 (1.16–2.99)	<0.001
Fetal Weight	0.43 (0.010–0.075)	<0.001	Fetal weight	0.027 (0.015–0.039)	<0.001

* OR: odd ratio; ^†^CI: confidence interval. The model was adjusted for maternal age, maternal body mass index, fetal weight and dilation time.

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
