# Peer review of "Influence of Body Mass Index on Gestation and Delivery in Nulliparous Women: A Cohort Study"

_ijerph, 2019, doi:10.3390/ijerph16112015_

Reviewer 1 Report

This manuscript covers an important topic of gestation and delivery complications associated with elevated BMI. Numerous gestational and delivery differences were found between normal weight and obese participants.

There are several areas that would benefit from additional explanation, as well as several typos.

Introduction:

(1) The introduction is quite choppy, with several paraphrase only having one sentence. Additional information on what results were found in the cited articles would do well to fill out this section and include additional context to the topic.

(2) It is not clear which specific analyses are novel. This should be explicitly stated.

Methods:

(3) It is not clear when data was collected from mothers. It is stated that this is a longitudinal design, but does not state at what points women were contacted or how data was collected. It is also stated that this is a retrospective study. Details into which aspects were longitudinal and which aspects of data collection were retrospective need to be included.

Results:

(4) There are several places in the results where both the obese and normal weight groups are referred to as group 1.

(5) In table 1 Fetal weight is said to be reported in grams, but then the averages are 3.369 and 3.251 so it is likely that they are in kg, but the SD seems to be reported in grams. Only one unit of measure should be used.

(6) In table 1 for fetal weight > 4000 g, it should be made clear that this line does not report fetal weight, but instead reports the number of children.

(7) Table 1 reports fetal weight > 4000 g and Table 2 reports fetal weight > 3999 g, unclear which is correct.

(8) Number of children born with a birth weight of > 4000 (or 3999) is reported in both Table 1 and Table 2. Unless there is a specific reason for the repetition, these data should only be reported in one table.

Discussion:

(9) The limitations section should be expanded. It is stated that there were limitations, but then only one is listed. In addition, the manuscript would benefit from having the one limitation listed expanded on. What might this imply about the results? Are there any other experimental studies that support your claim that obesity is causing the changes in the variables?

General:

(10) There are typos on lines 64 and 157-158.

Author Response

The authors would like to thank you for giving us the opportunity to revise and improve our manuscript; we also thank the reviewers for their thoughtful and constructive comments. We have considered the suggestions and have incorporated them into the revised manuscript. Changes the original manuscript are shown in red. We believe that our manuscript is stronger as a result of these modifications. An itemized point-by-point response to the reviewers’ comments is presented below.

Reviewer 1

This manuscript covers an important topic of gestation and delivery complications associated with elevated BMI. Numerous gestational and delivery differences were found between normal weight and obese participants.

There are several areas that would benefit from additional explanation, as well as several typos.

Introduction:

(1) The introduction is quite choppy, with several paraphrase only having one sentence. Additional information on what results were found in the cited articles would do well to fill out this section and include additional context to the topic.

We have added more information.

(2) It is not clear which specific analyses are novel. This should be explicitly stated.

We added it.

Methods:

(3) It is not clear when data was collected from mothers. It is stated that this is a longitudinal design, but does not state at what points women were contacted or how data was collected. It is also stated that this is a retrospective study. Details into which aspects were longitudinal and which aspects of data collection were retrospective need to be included.

The researchers collected data using the childbirth records belonging to the hospital. This register contained all information necessary for this study. The maternal weight and height of the women were recorded at the first consultation with the midwife. In Spain, prenatal control is free for all women, and the first visit to the midwife is at 8 weeks of gestation.

Results:

(4) There are several places in the results where both the obese and normal weight groups are referred to as group 1.

We have changed theses mistakes.

(5) In table 1 Fetal weight is said to be reported in grams, but then the averages are 3.369 and 3.251 so it is likely that they are in kg, but the SD seems to be reported in grams. Only one unit of measure should be used.

We have changed theses mistakes. We have used grams and no Kg.

(6) In table 1 for fetal weight > 4000 g, it should be made clear that this line does not report fetal weight, but instead reports the number of children.

We had added it.

(7) Table 1 reports fetal weight > 4000 g and Table 2 reports fetal weight > 3999 g, unclear which is correct.

The correct option is ≥4000g. We have changed it.

(8) Number of children born with a birth weight of > 4000 (or 3999) is reported in both Table 1 and Table 2. Unless there is a specific reason for the repetition, these data should only be reported in one table.

The table 2 shows the relative risks using a log binomial model.

Discussion:

(9) The limitations section should be expanded. It is stated that there were limitations, but then only one is listed. In addition, the manuscript would benefit from having the one limitation listed expanded on. What might this imply about the results? Are there any other experimental studies that support your claim that obesity is causing the changes in the variables?

We have added more limitations, and we have added update references that support ours results.

General:

(10) There are typos on lines 64 and 157-158.

We have changed it.

Reviewer 2 Report

This is a retrospective study performed in a regional hospital, which looks at the influence of BMI (body mass index) on a range of complications during gestation and delivery in nulliparous women. A strength of the study is that the authors performed power calculation and included the appropriate number of cases in each group. However, their main findings reiterate what has been shown already in previous studies. Moreover, many of the statistical analyses performed end up with very weak effects, which are difficult to interpret. The article is well focussed and easy to follow. However, some of the data presentation and analyses can be improved. Here are my main suggestions:

1.  It is not clear in the paper at which stage was the BMI for each case recorded. Prior to pregnancy, during late gestation (37-42 weeks), after delivery?

2.   I was surprised to see such a wide range of ages included in the study. The extreme ages (13-19 and ~40-43) may associate increased risks of complications regardless of the BMI status. Is this a wise choice? I wonder what would be the impact of excluding the extreme ages from the study. It’s also worth noting that the very crude BMI-based definition of obesity in adolescents is different than in adults. For children and adolescents between ages of 5 and 19: obese is BMI-for-age greater than 2 SD above the WHO median.

3.  The authors should make sure that there is consistency between data shown in tables and text. For example, the p value for the chi square test related to the increased risk of pre-eclampsia is mentioned as p=0.018 in the text (lane 100), but presented as p=0.049 in table 1. This lack of consistency triggered me to check it with a calculator and what I actually get is a p value that is marginally above 0.05 (p=0.07 for a Chi square with Yates correction. Note that the correct numbers to use are: 3, 107, 3, 607).

4.  The authors should also correct for multiple testing. This additional correction will surely eliminate some of the weaker p values.

5.  Correlations values are all very weak and their significance is perhaps questionable. For example a Pearson’s R=0.182 (their strongest correlation mentioned in the abstract) means that only ~3.3% of the variability observed (0.182 x 0.182) can be explained by this particular parameter. Perhaps a multivariate analysis would be more appropriate in finding a combination of parameters that are associated with a more robust increased risk for complications.

6.  Table 4 is difficult to follow. Each parameter listed in the first column (Age, BMI etc.) seems to have two rows of R values, but the meaning of each row is not explained. There are also some mistakes. For example, for the correlation between Age and BMI, the two values shown are 0.009 and 819(!) – the second value is probably 0.819 instead. Stars in the table are very difficult to observe.

7.  Throughout the text, strong p values should not be presented as “0.000”. Instead they could be shown using the “scientific format” (i.e. 0.0003 is 3.00E-04).

Given the very weak R values, the discussion should include a few phrases that mention the main strengths and limitations of this study.

Author Response

The authors would like to thank you for giving us the opportunity to revise and improve our manuscript; we also thank the reviewers for their thoughtful and constructive comments. We have considered the suggestions and have incorporated them into the revised manuscript. Changes the original manuscript are shown in red. We believe that our manuscript is stronger as a result of these modifications. An itemized point-by-point response to the reviewers’ comments is presented below.

Rieviewer 2:

This is a retrospective study performed in a regional hospital, which looks at the influence of BMI (body mass index) on a range of complications during gestation and delivery in nulliparous women. A strength of the study is that the authors performed power calculation and included the appropriate number of cases in each group. However, their main findings reiterate what has been shown already in previous studies. Moreover, many of the statistical analyses performed end up with very weak effects, which are difficult to interpret. The article is well focussed and easy to follow. However, some of the data presentation and analyses can be improved. Here are my main suggestions:

1. It is not clear in the paper at which stage was the BMI for each case recorded. Prior to pregnancy, during late gestation (37-42 weeks), after delivery?

The researchers collected data using the childbirth records belonging to the hospital. This register contained all information necessary for this study. The maternal weight and height of the women were recorder at the first prenatal consultation with the midwife. In Spain, prenatal control is free for all women and the first visit to a midwife is at 8 weeks of gestation.

2. I was surprised to see such a wide range of ages included in the study. The extreme ages (13-19 and ~40-43) may associate increased risks of complications regardless of the BMI status. Is this a wise choice? I wonder what would be the impact of excluding the extreme ages from the study. It’s also worth noting that the very crude BMI-based definition of obesity in adolescents is different than in adults. For children and adolescents between ages of 5 and 19: obese is BMI-for-age greater than 2 SD above the WHO median.

In Spain is frequent that women with 40-43 age are pregnancy, but we have retired the women with extreme age (younger than 18 ages and older than 40).  

3. The authors should make sure that there is consistency between data shown in tables and text. For example, the p value for the chi square test related to the increased risk of pre-eclampsia is mentioned as p=0.018 in the text (lane 100), but presented as p=0.049 in table 1. This lack of consistency triggered me to check it with a calculator and what I actually get is a p value that is marginally above 0.05 (p=0.07 for a Chi square with Yates correction. Note that the correct numbers to use are: 3, 107, 3, 607).

We had used the Fisher´s test for preclampsia, because is a robust test when there are less than 5. In the table the dates were corrects, but in the text we did a mistake.

4. The authors should also correct for multiple testing. This additional correction will surely eliminate some of the weaker p values.

We have correct all test.

5. Correlations values are all very weak and their significance is perhaps questionable. For example a Pearson’s R=0.182 (their strongest correlation mentioned in the abstract) means that only ~3.3% of the variability observed (0.182 x 0.182) can be explained by this particular parameter. Perhaps a multivariate analysis would be more appropriate in finding a combination of parameters that are associated with a more robust increased risk for complications.

We have used multivariate linear regression model with the quantitative variables that was significance in Pearson´s R (maternal age,BMI, fetal weight and dilatation time).

6. Table 4 is difficult to follow. Each parameter listed in the first column (Age, BMI etc.) seems to have two rows of R values, but the meaning of each row is not explained. There are also some mistakes. For example, for the correlation between Age and BMI, the two values shown are 0.009 and 819(!) – the second value is probably 0.819 instead. Stars in the table are very difficult to observe.

We have decided to remove this table. It is true, that correlation values are very weak and we have done a logistic regression model.

We have added a new table with multivariate linear regression model with associated factors and dilation and expulsion times

7. Throughout the text, strong p values should not be presented as “0.000”. Instead they could be shown using the “scientific format” (i.e. 0.0003 is 3.00E-04).

We had changed p=0.000 for p<0.001< span="">.

Given the very weak R values, the discussion should include a few phrases that mention the main strengths and limitations of this study.

We have included more limitations of this study.

Reviewer 3 Report

Rodriguez-Mesa et co-workers aimed to assess the influence of obesity on pregnancy and childbirth in pregnant nulliparous women. They revised the obstetric history of 720 women, splitted into two subgroups according to the deficion of normoweight women (BMI<25) and="" obesity="" bmi="">30). They found longer dilation time in obese women than in normoweight women. Obese women were more likely to have gestational diabetes (relative risk (RR)=3.633, 95% CI 2.114-6.245, P=0.000), pre-eclampsia (RR=5.545, 95% CI 27 1.134-27.122, P=0.018), induced birth (RR=1.262, 95% CI 1.061-1.502, P=0.016) and caesarean section (RR=2.103, 95% CI 1.083-4.085, P=.0391) than normoweight women. They concluded that obesity is associated with increased complications during pregnancy, including highr cesarean section and labour induction rates.

The paper in not original, so it does not enrich the current literature on this specific topic. The aim of the abstract includes aspects missed (i.e. childbirth) in the paper. The Authors identified only two categories of BMI, without consider the overweight condition. Improvements in the statistical analysis are required (logistic regression in place of Pearson correlation). References must be updated. English revision is necessary.

Author Response

The authors would like to thank you for giving us the opportunity to revise and improve our manuscript; we also thank the reviewers for their thoughtful and constructive comments. We have considered the suggestions and have incorporated them into the revised manuscript. Changes the original manuscript are shown in red. We believe that our manuscript is stronger as a result of these modifications. An itemized point-by-point response to the reviewers’ comments is presented below.

(1) Improvements in the statistical analysis are required (logistic regression in place of Pearson correlation).
(2) References must be updated.
(3) English revision is necessary.

(1) We have added a new table with multivariate linear regression model with associated factors and dilation and expulsion times
(2) We have updated references.
(3) We have send our manuscript to edit to American Journal Expert Services.

Round  2

Reviewer 1 Report

There have been extensive improvements to this manuscript. These changes enhance the quality and readership of the text.The introduction is still light on background information, but all requested revision have been completed.

Reviewer 2 Report

The authors dealt reasonably well with all my previous comments. I have no further suggestions.